# Validity of self-reported weight and height among female young adults in the United Arab Emirates

Dalia Haroun[ID]*, Aseel Ehsanallah

College of Natural and Health Sciences, Department of Public Health and Nutrition, Zayed University, Dubai, United Arab Emirates

* Dalia.haroun@zu.ac.ae

## Abstract

Self-reported weight and height serve as important metrics in estimating overweight and obesity prevalence within epidemiological studies, primarily due to their cost and time efficiency. However, the accuracy and reliability of these self-reported measures remain controversial, with conflicting reports emerging from different regions. This study aims to compare self-reported weight and height with measured values among young female adults in the United Arab Emirates. A cross-sectional study of 131 female university students aged 17–27 reported their weight and height on a self-administered questionnaire and on the same day had their height and weight measured. Body Mass Index (BMI) values of both self-reported and measured weight and height were calculated and categorized according to the World Health Organization's cut-off points. Overall, 87% of students had a resultant self-reported BMI value within their actual BMI category. The mean differences between self-reported and measured weight and height in the present study were -0.92 kg and 0.38 cm, respectively. Results indicated strong agreement between self-reported and direct measurements, as demonstrated by weighted Kappa statistics (kappa = 0.87). Bland & Altman plots illustrated that the majority of values fell within the limits of agreement (2 SD), with no systemic bias detected. BMI calculated from self-reported data demonstrates high sensitivity and specificity. Linear regression analyses revealed that self-reported weight ($r^2$ = 0.973; p<0.001), height ($r^2$ = 0.902; p<0.001), and BMI ($r^2$ = 0.964; p<0.001) accurately predicted measured weight, height, and BMI. The study's results highlight the ability of female university students in the UAE to accurately provide self-reports of their weight and height. This finding provides further support for the utilization of self-reported data on height and weight as a valid method for collecting anthropometric information.

## Introduction

In the United Arab Emirates (UAE), overweight and obesity are major public health concerns and are key contributors to chronic illnesses including Type II diabetes, cardiovascular, respiratory, and gallbladder diseases, and some types of cancers [1, 2]. Addressing these challenges

**Data Availability Statement:** All relevant data are within the manuscript.

**Funding:** The author(s) received no specific funding for this work.

**Competing interests:** The authors have declared that no competing interests exist.

requires a foundation of reliable data which is essential for designing effective interventions, conducting precise risk assessments, and developing specific healthcare strategies. Accurate information not only aids in the development of preventive measures but also leads to improved health outcomes.

The "gold standard" in obtaining accurate results of weight and height is to have appropriately trained and monitored personnel who perform direct measurements of these anthropometrics using standardized and well-managed equipment and methods [3]. However, the application of this "gold standard" is not feasible, particularly in large-scale epidemiological studies, which commonly use self-reported weight and height data due to constraints such as time, financial resources, and available personnel [4–6]. Therefore, healthcare practitioners utilize self-reported height and weight to calculate BMI, providing a consistent method for assessing obesity and overweight trends in populations [7]. Such a method allows for quick, easy, and convenient data collection that can be completed through face-to-face or telephone interviews or self-administered questionnaires at a minimal cost and resources, especially for large-scale studies [1, 3, 8, 9]. Policymakers rely on this information to allocate resources and establish healthcare priorities, emphasizing the need to evaluate its precision and reliability [4]. These anthropometric measurements serve as primary factors of investigation and potential variables that might introduce confounding influences. They are fundamental in nutritional status assessments, predicting functional limitations, disease risks, and overall mortality [10]. Despite the inherent challenges, the use of self-reported data remains integral to understanding public health patterns and informing healthcare strategies.

The use of self-reported data in research is questionable and it can introduce limitations related to recall bias of participants who overestimate or underestimate their weight or height [1, 9] or who simply cannot recall their actual weight or height [11]. Bias in self-reporting can result in inaccuracies when evaluating nutritional status, ultimately compromising the precise evaluation of overweight/obesity prevalence within a community [12]. The literature shows substantial differences between the subjectively and objectively determined BMI. The subjective BMI tends to underestimate the objective BMI which consequently results in the underestimation of the prevalence of overweight and obesity [13, 14]. The lack of concordance between the subjectively and objectively determined BMI is due to the fact that weight tends to be under-reported, and height is often over-reported [8, 11, 15, 16]. Variations in the accuracy of self-reported weight and height among populations depend on some factors including age, gender, weight status, race [9, 15], and cultural factors [17]. There is a tendency of some individuals with these different factors to report weight and height values that are idealistic from their own or society's perspective [18].

Cultural factors and backgrounds can play a role in the variations between self-reported and measured weight and height. A study of the European Union that focused on the relationship between the subjective and objective BMI among the European countries found that the degree of correlation between the subjective and objective BMI differed from one country to another thus, comparable estimates of the prevalence of overweight and obesity could not be made based on the subjective BMI [19]. This difference between the actual BMI and the perceived one can be a result of different views and perceptions on beauty and ideal body image that vary from one culture to another, leading individuals to misreport data in an attempt to attain a culturally valued body image [14]. Cultural factors and race go hand in hand because different ethnic groups have different cultural perceptions on ideal weight or height which may influence their tendencies to report data accurately. For example, white individuals are more likely to overestimate their height than individuals from other ethnic groups including black and Hispanic individuals [20]. Evaluating the accuracy of self-reported weight and height values requires a direct comparison between the self-reported data and the measured values

within the target population. This step is crucial to determine the extent of the biases among populations, influenced by cultural norms and societal factors [12, 21].

While young adulthood is typically seen as a period associated with peak health and well-being, recent data reveals a notable change in the distribution of BMI. This shift is marked by a decline in the proportion of individuals falling into the 'normal' BMI category and a simultaneous increase in those categorized as 'overweight and obese' [22]. Adolescents are more likely to report their weight and height inaccurately compared to adults due to the rapid growth period they are undergoing that leads to substantial physical changes. Thus, they tend to lack knowledge about their current weight and height [14, 15]. Some aspects may influence the accuracy of self-reported data among adolescents, these aspects include body image and social desirability which both can lead to reporting values that are considered ideal or socially acceptable [23].

In both women and individuals struggling with overweight and obesity, there is a common tendency to underestimate their actual body weight [22, 24]. Overweight and obese individuals usually underreport their weight compared to underweight and normal-weight individuals [1, 9, 25]. This may be because of the stigma that is attached to being heavy which leads those who are overweight or obese to underestimate their actual weight [8]. Females are more likely than males to underestimate their weight [1, 3, 9, 26, 27]. This may be attributable to the role of media and advertisement that highlight women's status mostly on the basis of their appearance which may influence females to report data that is desired by society [28]. Underestimation of weight among males is applicable only to those who are overweight or obese [28]. Males, in general, are more likely to overestimate their height than females [1]. Societal pressures and media influence lead many, especially women and overweight individuals, to underestimate their weight which can compromise the validity of self-reported anthropometric measurements.

The topic of the relationship between self-reported and measured weight, height, and BMI has been studied internationally including the United States and several European countries, but the literature shows no results regarding the UAE. Hence, this study aims to assess the validity of self-reported anthropometric measurements (weight and height) and BMI classification amongst female university students in the UAE.

## Materials and methods

Ethical approval was obtained from the research ethics committee at Zayed University (ZU15_101_F). This research was performed as a part of a cross-sectional study investigating the caffeine and energy drink consumption among university students in the UAE. Data was collected between 25th February and 17th March 2016. Convenience sampling was used. The sample size was not calculated for this study as it was a post-hoc analysis. Based on results from a similar study [29], using 0.55kg as the mean weight difference and 2.03kg as the SD of the difference a sample size of 109 would be needed to achieve a power of 80% at 0.05 level of significance [30]. Our sample size of 131 is therefore comparable to what was used in similar studies [22].

All female students studying at Zayed University in Dubai were eligible to participate in the study. Those who were pregnant or had electronic medical implants were excluded. Researchers explained the study to participants and written informed consent was obtained prior to the study.

First, participants self-completed the questionnaire, that was available in both Arabic and English language. The questionnaire was used to obtain demographic data (age, ethnicity, college, year of study). Participants were also asked to report their weight in kilograms and height in centimeters. Subsequently, participants' weight and height were measured by trained researchers, holding an undergraduate degree in Public Health and Nutrition, according to

standard protocol. Participants were instructed to remove their shoes and socks and were requested to remove any heavy objects from their pockets (e. g. mobile phones, keys, key chains, wallets, and heavy accessories). Height was measured standing upright facing forward with back, buttocks, and heels vertically aligned against the scale. Additionally, feet and heels were placed together, and the movable head plate rested firmly on the top of participants' crown. Height was measured to the nearest 0.1cm using the portable stadiometer (Charder, HM-200P) and was set up on a flat, secure, stable surface against a wall. Weight was measured to the nearest 0.1kg using the portable Tanita Body Composition Analyser (BC-420MA). This study aims to assess the validity of self-reported anthropometric measurements (weight and height) and BMI classifications amongst female university students in the UAE.

## Data analysis

Self-reported and measured BMI were calculated as weight in kilograms divided by height in meters squared ($kg/m^2$). Participants were classified as either underweight ($<18.5$ $kg/m^2$), healthy weight 18.5–24.9 $kg/m^2$, overweight 25–29.9 $kg/m^2$, or obese $\geq 30$ $kg/m^2$ using World Health Organization cut-off points [31]. Descriptive statistics were used to analyze demographic data. Means and SD for weight, height, and BMI were computed for self-reported and measured data. Paired sample t-tests were used to determine the differences between self-reported and measured anthropometrics for the whole sample and then stratified by BMI category. The mean difference was calculated as self-reported values minus measured values. Correlations between the methods were tested using Pearson Correlations. Bland-Altman plots were performed to assess the agreement between self-reported and directly measured weight and height [32]. Means of self-reported and measured values were computed. The differences between self-reported and measured values were plotted against their means with a mean difference plus or minus 1.96 times its standard deviation. The Interclass Correlation Coefficient (ICC) was used to derive a summary measure of absolute agreement between self-reported and measured weight and height, where ICC $>0.75$ indicates good reliability [33]. Kappa statistics were calculated to assess the degree of agreement between BMI categorization derived from self-reported data versus that derived from measured data, where a kappa $>0.8$ indicates a strong strength of agreement [34, 35]. The effectiveness of self-reported weight and height data in accurately identifying underweight, overweight, and obesity was assessed through various measures, including sensitivity, specificity, positive predictive values (PPV), and negative predictive values (NPV). Sensitivity determines how accurately self-reported data identify individuals with underweight, overweight, or obesity, while specificity measures how accurately it excludes those without these conditions. PPV indicates the proportion of reported cases confirmed, while NPV shows the proportion of non-reported cases confirmed [15]. Linear regression analyses (adjusted for age) were used to assess the accuracy of self-reported weight, height, and BMI in predicting measured values. Data was analyzed with SPSS software, version 29 for Windows (IBM, Armonk, NY, USA) and significance was set at p-values $<0.05$.

## Results

A total of 131 female participants aged between 19 and 27 years provided self-reports of their weight and height, and had their measurements taken. The mean age (standard deviation) of participants was 19.7 (1.9) years. The majority of participants were Emirati (93.9%). The sample was evenly distributed across the different years of study and came from a variety of colleges. Additional information on participants' demographics can be found in Table 1.

Approximately half the participants (42.7%) were healthy weight, 22.9% overweight and 14.5% obese. Self-reported and measured anthropometric data were significantly correlated

**Table 1. Characteristics of the study participants (n = 131).**

| Variables | n (%) |
|---|---|
| **Age** | |
| 17–19 | 68 (51.9) |
| 20–27 | 63 (48.1) |
| **Nationality** | |
| Emirati | 124 (93.9) |
| Non-Emirati | 5 (3.8) |
| Missing | 3 (2.3) |
| **Marital status** | |
| Single | 116 (88.5) |
| Engaged | 6 (4.6) |
| Married | 8 (6.1) |
| Divorced | 1 (0.8) |
| **Has children** | |
| Yes | 2 (1.5) |
| No | 129 (98.5) |
| **Year of study** | |
| Foundation year | 23 (17.6) |
| 1st year | 34 (26.0) |
| 2nd year | 17 (13.0) |
| 3rd year | 33 (25.2) |
| 4th year | 24 (18.3) |
| **College** | |
| Academic Bridge Program | 23 (17.6) |
| University College | 43 (32.8) |
| Art and Creative Enterprises | 5 (3.8) |
| Business Sciences | 5 (3.8) |
| Communication and Media Science | 11 (8.4) |
| Education | 4 (3.1) |
| Sustainability Sciences and Humanities | 28 (21.4) |
| Technological Innovation | 10 (7.6) |
| Missing | 2 (1.5) |
| **Took nutrition courses** | |
| Yes | 33 (25.2) |
| No | 98 (74.8) |

with weight (r = 0.997; p<0.001) height (r = 0.988; p<0.001) and BMI (r = 0.996; p<0.001). Furthermore, there was a strong agreement between self-reported values for height, weight, and the calculated BMI, where ICC values were 0.848, 0.982, and 0976 respectively (p<0.001).

Table 2 presents the number and proportion of female participants (n = 131) categorized into different BMI classifications based on both self-reported and measured values of height and weight. The data reveals distinct patterns in participants' self-perception of their weight

**Table 2. The number and proportion of female participants (n = 131) were categorized into different BMI classifications based on both self-reported and measured values of height and weight.**

| | | BMI-category based on Measured BMI | | | | | |
|---|---|---|---|---|---|---|---|
| | | Underweight n (%) | Normal weight n (%) | Overweight n (%) | Obese n (%) | Total n (%) | Kappa (95% CI) |
| **BMI-category based on Self-reported BMI** | Underweight | 21 (16) | 2 (1.5) | 0 (0.0) | 0 (0.0) | 23 (17.6) | 0.87 (0.81, -0.93) |
| | Normal weight | 5 (3.8) | 53 (40.5) | 7 (5.3) | 0 (0.0) | 65 (49.6) | |
| | Overweight | 0 (0.0) | 1 (0.8) | 23 (17.6) | 2 (1.5) | 26 (19.8) | |
| | Obese | 0 (0.0) | 0 (0.0) | 0 (0.0) | 17 (12.9) | 17 (12.9) | |
| | Total | 26 (19.8) | 56 (42.7) | 30 (22.9) | 19 (14.4) | 131 (100) | |

status compared to their actual measured BMI categories. According to the measured data, 19.8% of participants were classified as underweight, with a slight under-reporting observed as only 17.6% self-reported being underweight. In the normal weight category, 42.7% of participants were categorized based on measured BMI, whereas 49.6% self-identified as normal weight, indicating a tendency to perceive oneself as normal weight despite measured differences. Notably, 22.9% were classified as overweight using measured BMI, contrasting with the 19.8% who self-reported being overweight, indicating underreporting among this group. In the obese category, 14.4% of participants were classified based on measured BMI versus 12.9% based on self-reported measurements. The high Kappa value of 0.87 (95% CI: 0.81–0.93) demonstrated a statistically significant strong agreement between self-reported and measured BMI categories among the participants.

Table 3 illustrates the mean difference between self-reported versus measured weight, height, and BMI of the total sample (n = 131). Overall, participants significantly underreported their weight by 0.92 kg (p = 0.001). Height was significantly over-reported by an average of 0.38 cm (p = 0.013). Underestimating weight and overestimating height resulted in a significant underestimation of BMI by an average of 0.47 kg/m$^2$ (p<0.001).

Table 4 demonstrates the difference between self-reported versus measured weight, height, and BMI by BMI category. Underweight students significantly over-reported weight by 0.50 kg (p = 0.033). Normal-weight students significantly underreported weight by 0.51 kg (p = 0.044), consequently BMI was significantly under-estimated by 0.28 kg/m$^2$ (p = 0.029). Overweight students underestimated weight by 1.05 kg (p = 0.071) but significantly over-reported height the most by 0.74 cm (p = 0.044) resulting in a significant underestimation of BMI by 0.64 kg/m$^2$ (p = 0.007). Obese students significantly underreported weight and BMI by 3.89 kg (p = 0.004) and 1.65 kg/m$^2$ (0.002), respectively.

Table 5 summarizes the diagnostic values of self-reported height and weight to determine underweight, normal weight, overweight, and obesity among female participants. The results demonstrate high sensitivity and specificity of self-reported BMI compared to measured BMI. The sensitivity for the overweight category was slightly lower at 76.7% with a specificity of 97%, compared to the obese category which exhibited higher sensitivity and specificity (89.5% and 100%, respectively). Moreover, the PPV, was 88.5% for overweight and 100% for obesity, representing the proportion of females that correctly reported their anthropometric measures. The corresponding NPV for overweight and obesity were 93.3% and 98.2%, respectively, indicating the proportion of non-reported cases confirmed.

The agreement between self-reported and directly measured weight, height, and BMI at an individual level is illustrated graphically in the Bland and Altman plots (Fig 1). The 95% limits of agreement (LOA) for weight (+4.98 to -6.85), height (+3.80 to -3.03), and BMI (+2.01 to

**Table 3. Self-reported versus measured anthropometrics in all participants (n = 131).**

| | Self-reported (mean ±SD) | Measured (mean ±SD) | Mean difference* (95% CI) | P-value |
|---|---|---|---|---|
| **Weight (kg)** | 59.5 ±15.8 | 60.4 ±17.1 | -0.92 (-1.45 to -0.40) | 0.001 |
| **Height (cm)** | 158.7 ±5.4 | 158.3 ±5.5 | 0.38 (0.08 to 0.69) | 0.013 |
| **BMI (kg/$m^2$)** | 23.5 ±5.9 | 24 ±6.4 | -0.47 (-0.69 to -0.25) | <0.001 |

* Mean difference = self-reported—measured values.

BMI (Body Mass Index); SD (Standard Deviation); CI (Confidence Interval)

**Table 4. Agreement between self-reported and measured values for height, weight, and BMI stratified by BMI category.**

| | | Self-reported (mean ±SD) | Measured (mean ±SD) | Mean difference* (95% CI) | P-value |
|---|---|---|---|---|---|
| **Underweight 26 (19.8%)** | Weight (kg) | 41.8 ±4.2 | 41.3 ±3.9 | 0.50 (0.04 to 0.96) | 0.033 |
| | Height (cm) | 156.1 ±4.8 | 156.1 ±5.0 | 0.05 (-0.52 to 0.61) | 0.868 |
| | BMI (kg/$m^2$) | 17.1 ±1.3 | 16.9 ±1.1 | 0.19 (-0.05 to 0.42) | 0.110 |
| **Normal weight 56 (42.7%)** | Weight (kg) | 54.3 ±6.2 | 54.8 ±6.5 | -0.51 (-1.00 to -0.01) | 0.044 |
| | Height (cm) | 159.2 ±5.3 | 158.8 ±5.6 | 0.34 (-0.23 to 0.92) | 0.237 |
| | BMI (kg/$m^2$) | 21.4 ±1.9 | 21.7 ±1.9 | -0.28 (-0.53 to -0.03) | 0.029 |
| **Overweight 30 (22.9%)** | Weight (kg) | 66.7 ±6.0 | 67.7 ±5.7 | -1.05 (-2.20 to 0.09) | 0.071 |
| | Height (cm) | 159.8 ±5.1 | 159.0 ±5.2 | 0.74 (0.26 to 1.22) | 0.004 |
| | BMI (kg/$m^2$) | 26.1 ±1.7 | 26.7 ±1.1 | -0.64 (-1.10 to -0.19) | 0.007 |
| **Obese 19 (14.5%)** | Weight (kg) | 87.4 ±12.3 | 91.3 ±13.9 | -3.89 (-6.33 to -1.45) | 0.004 |
| | Height (cm) | 159.2 ±6.2 | 158.7 ±6.3 | 0.41 (-0.27 to 1.09) | 0.222 |
| | BMI (kg/$m^2$) | 34.5 ±4.9 | 36.2 ±5.0 | -1.65 (-2.59 to -0.71) | 0.002 |

* Mean difference = self-reported—measured values.

-2.95) were quite far from zero indicating an overall discrepancy between self-reported and measured values. Some participants had an extreme difference between self-reported and measured anthropometrics that fell outside the 95% LOA. There were five values that under-reported weight by more than 10 kg most of which were near 100 kg. There was one value that over-reported height by more than 10 cm and five values that under-reported BMI by more than 4 kg/m² most of which were in the obese range. The variability of the difference increases for larger weight and BMI, however, this was observed only for a small number of individuals. Therefore, excluding those outliers we can see from the graph that there was a good overall agreement between individual measures.

Additionally, linear regressions analyses showed that self-reported weight, height, and BMI were accurate in predicting measured weight ($r^2$ = 0.973; p<0.001), height ($r^2$ = 0.902; p<0.001), and BMI ($r^2$ = 0.964; p<0.001). Furthermore, there was no systematic bias observed over the range of measurements for weight (r = 0.451; p <0.001) height (r = 0.065; p = 0.458), or BMI (r = 0.387; p <0.001).

**Table 5. Sensitivity, specificity, positive predictive value, and negative predictive value of self-reported BMI classifications.**

| | Sensitivity (%) | Specificity (%) | PPV (%) | NPV (%) |
|---|---|---|---|---|
| **Underweight** | 80.8 | 98 | 91.3 | 95.4 |
| **Normal weight** | 94.6 | 84 | 81.5 | 95.5 |
| **Overweight** | 76.7 | 97 | 88.5 | 93.3 |
| **Obesity** | 89.5 | 100 | 100 | 98.2 |

a)
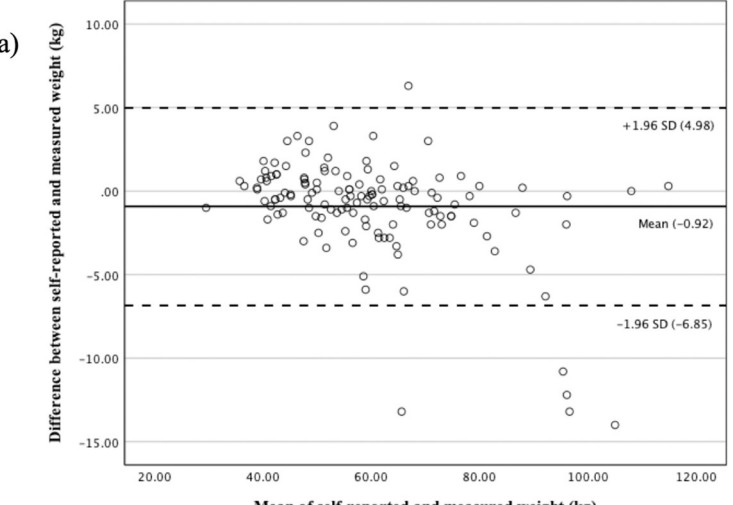

b)
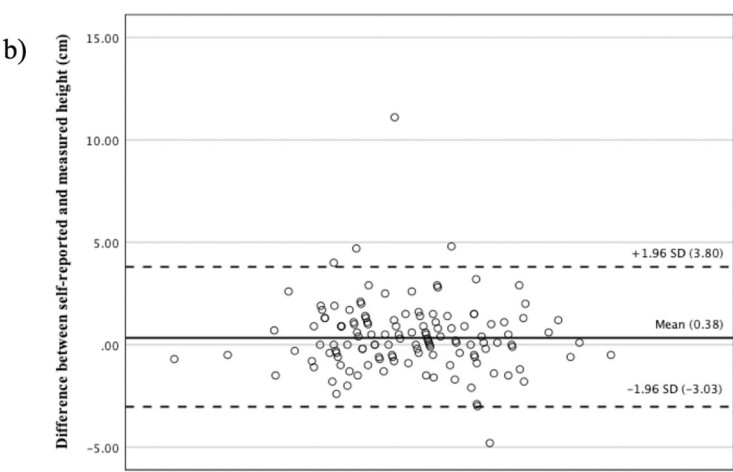

c)
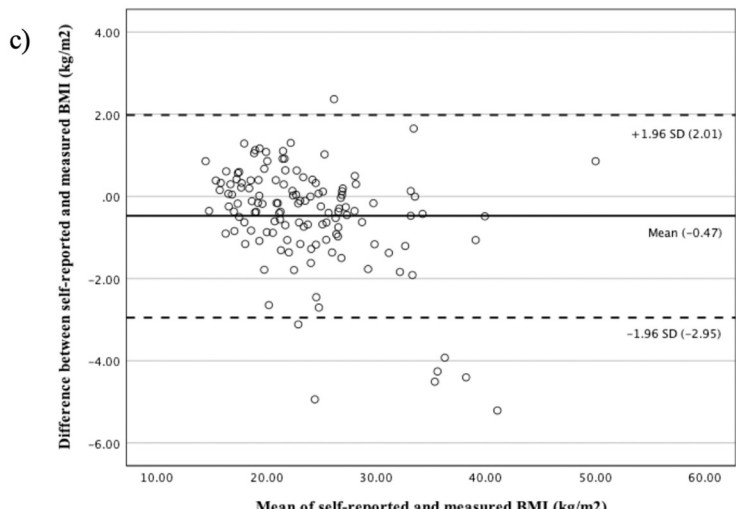

**Fig 1.** Bland and Altman plot (B&A) of self-reported versus measured (a) weight, (b) height, and (c) BMI. Dark line: Mean difference between self-reported and measured anthropometrics. Dotted line: 95% limits of agreement (LOA), in which the upper line is +1.96 SD and lower line is -1.96 SD from mean difference (red line).

## Discussion

The present study was the first to be carried out in the UAE. It examined the difference between self-reported and measured weight, height, and BMI generally and specifically by weight status in a total of 131 young female students.

The findings presented in this study emphasize the complexity of body image perception and its impact on self-reported and measured BMI categories among young female participants. The substantial agreement observed between self-reported and measured BMI categories, as indicated by a high Kappa value of 0.87 (95% CI: 0.81–0.93), suggests a strong alignment between participants' self-perception and their actual weight status. This level of agreement is crucial in understanding the accuracy of self-reported data. The study reveals interesting patterns of self-perception across different BMI categories. While a slight under-reporting was observed in the underweight category, participants showed a tendency to perceive themselves as normal weight, even when their measured BMI suggested otherwise. Notably, there was a consistent trend of under-reporting in the overweight and obese categories.

The general trend of weight under-reporting and height over-reporting found in this study is consistent with other studies carried out among female college students, yet the mean differences vary. The mean differences between self-reported and measured weight and height in the present study were -0.92 kg and 0.38 cm, respectively. Weekly body weight fluctuations are around 0.35% (equivalent to 0.2 kg in this study) [36]. Quick et al. (2015) found mean differences of -0.27 kg for weight and 0.51 cm for height among female students from eight universities in the U.S., which is lower for weight and larger for height compared to findings from this study [37]. Even higher differences were seen among female college students in Italy as weight was under-reported by 1.9 kg and height was over-reported by 2.8 cm [16]. A possible reason for the inconsistency of mean differences is the diverse cultural perspectives of ideal body size that can affect the extent of weight underestimation and height overestimation from one region to another as discussed earlier in the introduction [14].

The results of this study found that self-reported BMI demonstrates high sensitivity and specificity, which suggests that self-reported height and weight can effectively classify individuals into the different BMI categories, with reasonably accurate results. High sensitivity (89.5%) and specificity (100%) values for obesity were observed in this study. These findings closely resemble those of a previous study by Lee et al. (2011), which reported a sensitivity of 83.6% and a specificity of 98% for the prevalence of obesity [10].

The Bland & Altman plots illustrated limited disparities between self-reported weight, height, and BMI at an individual level. Estimations for body weight, height, and BMI were found to be within the pre-defined limits of accuracy, indicating that self-reported measurements can be considered a reliable tool for estimating a person's weight, height, and BMI in this sample population.

This study found that normal-weight, overweight, and obese students underreported their weight. The underestimation increased as weight increased which implies that obese and over-weight students under-reported weight to a higher extent than normal-weight students. In comparison with prior research among female college students by Gunnare et al. (2013) in the U.S. and Larsen et al. (2008) in the Netherlands weight was under-reported only among over-weight and obese subjects, but not among those who were normal weight [8, 26]. A possible reason for this might be that these normal-weight students in the U.S. and Netherlands were

more health-aware compared to the participants in this study. This can be alarming and a possible sign of an eating disorder or extreme dieting because despite being within a healthy weight range, these students failed to recognize their own health status. Additionally, these normal-weight students might have thought that they needed to lose weight, thus underreporting their weight. Concerning obese and overweight students, it is speculated that their stronger inclination toward thinness could be influenced by societal and media standards to a greater extent than normal-weight students. Notably, a study has linked weight under-reporting among heavier subjects to depression. Sherry, et al. (2007) pointed out that heavier individuals were more prone to depression, leading them to under-report their weight [9]. Another probable interpretation for weight underestimation among overweight and obese students could be their reluctance to acknowledge their heavy-weight status. If this is indeed the case, it is concerning, as it implies a lack of awareness about the health risks associated with excess weight. Moreover, it raises questions about their willingness to adopt necessary dietary and lifestyle changes to manage their weight effectively.

The present study is the first among female college students to report a slight weight overestimation of underweight participants. Only one study in Sweden assessing adolescents with a mean age of 16 years supported this finding [23]. Over-reporting of weight amongst underweight students might be a positive indicator of acknowledging that their thinness was unhealthy, and their willingness to be in a healthy weight range. On the contrary, weight over-reporting could have a negative meaning just in the case of eating disorders particularly anorexia, in which individuals who are extremely thin perceive themselves as heavy.

Variations in height were observed among students of different weight statuses, with individuals classified as heavyweight showing the highest tendency to over-report their height. Earlier studies amongst female college students have not examined the relationship between self-reported and measured height by BMI categories. The sole exception was the Swedish adolescent study, which demonstrated an increasing trend in height overestimation with higher BMI, aligning with the results of this study [23]. This implies that overweight and obese students might be more inclined to overstate their height, potentially as a way to compensate for their excess weight and avoid appearing as heavy as their actual weight suggests.

The outcomes suggest a high percentage of correct BMI classification when relying on self-reported weight and height, with only 18 out of 131 (13.7%) individuals being misclassified. In comparison, Lasren, et al. (2008) found a larger percentage of BMI misclassification when using self-reports as 50% of overweight and obese female college students in the Netherlands were misclassified as normal weight [8]. A possible reason for the inconsistency of BMI misclassification may be due to sample size as Larsen's et al., study included 209 students while this study included 131 students.

The findings of this study suggest that self-reported values can be considered when determining the prevalence of unhealthy body weight for targeting weight loss or gain interventions. Female university students in the UAE were generally able to provide self-reports of their weight and height. While self-reported data showed small discrepancies from measured values, with weight and BMI tending to be underestimated and height overestimated, the overall trend indicated that self-reports could still be utilized. It is important to note that the accuracy of self-reports decreased with higher BMI, leading to a skewed prevalence of overweight and obesity. Given the convenience and minimal cost of self-reports, they will continue to be utilized, particularly for large-scale studies. However, it is recommended to prioritize direct measurements whenever feasible. Strategies to minimize self-reporting errors include employing a two-method measurement approach, involving direct measurements for a small portion of the sample, to estimate accuracy for the entire population. This method allows researchers to estimate the accuracy of values for the entire sample, reducing bias [13]. Additionally,

participants can be encouraged, if possible, to measure themselves before completing self-administered questionnaires for enhanced accuracy [15]. Ultimately, to increase the reliability of self-reported data, it is important to establish a mandatory body size surveillance system for students, which periodically screens for cases of underweight, overweight, and obesity among college students, fostering greater awareness of weight and height among the student population.

The results stress the need for several intervention programs at Zayed University in Dubai including overweight and obesity controlling programs, programs for weight gain of under-weight students as well as screening and intervention programs for potential eating disorders. The efforts of these programs should be directed to sustainable and easy-to-follow changes including dietary, physical activity, and behavioral modifications. Needless to say, all of these changes will require an effective team of dieticians, health counselors, and educators who can help in one-to-one sessions as well as deliver messages to the students as a whole.

A limitation of this study is that the sample size was not calculated a priori, as it was a post-hoc analysis. This may have impacted the statistical power and generalizability of the findings. The generalizability of the current study may also be limited by the recruitment of a convenience sample of only female participants. The absence of male participants may limit the generalizability of the findings to the broader population. A strength of this study is the absence of a time gap between self-reported data and direct measurements, effectively reducing the potential for weight fluctuations often seen in young adults [37]. Future research should examine the validity of self-reported measures among male participants and examine the relationship between self-reported and measured weight height, and BMI by gender and weighing frequency amongst college students from other emirates in the UAE. Additionally, further research is needed to determine the extent to which self-reporting might change in the later stages of adulthood. Lastly, it would be interesting to study this topic among the Arab countries to determine cultural and race variations in self-reports.

## Conclusion

This study among female university students in Dubai offers valuable insights into the complex dynamics of body image perception and its impact on self-reported and measured weight, height, and BMI. The findings underscore the importance of precise measurements and individualized interventions to address weight-related concerns effectively. These insights are essential for intervention programs, emphasizing the need for tailored initiatives focused on obesity prevention, healthy weight gain, and interventions for potential eating disorders. Additionally, the study's outcomes reveal a high level of agreement between self-reported and measured data, highlighting the reliability of self-perception in determining weight status among this sample of young female adults. This finding provides further support for the utilization of self-reported data on height and weight as a valid method for collecting anthropometric information when direct measurements are not possible.

## Author Contributions

**Conceptualization:** Dalia Haroun.

**Formal analysis:** Dalia Haroun, Aseel Ehsanallah.

**Project administration:** Dalia Haroun.

**Supervision:** Dalia Haroun.

**Writing – original draft:** Dalia Haroun.

**Writing – review & editing:** Dalia Haroun, Aseel Ehsanallah.

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
