## [Decision Letter · Decision Letter 0]

23 Feb 2024

PONE-D-23-35645Validity of self-reported weight and height among female young adults in the United Arab EmiratesPLOS ONE

Dear Dr. Haroun,

Thank you for submitting your manuscript to PLOS ONE. After careful consideration, we feel that it has merit but does not fully meet PLOS ONE’s publication criteria as it currently stands. Therefore, we invite you to submit a revised version of the manuscript that addresses the points raised during the review process. Please submit your revised manuscript by Apr 08 2024 11:59PM. If you will need more time than this to complete your revisions, please reply to this message or contact the journal office at plosone@plos.org. Please include the following items when submitting your revised manuscript:A rebuttal letter that responds to each point raised by the academic editor and reviewer(s). You should upload this letter as a separate file labeled 'Response to Reviewers'.A marked-up copy of your manuscript that highlights changes made to the original version. You should upload this as a separate file labeled 'Revised Manuscript with Track Changes'.An unmarked version of your revised paper without tracked changes. You should upload this as a separate file labeled 'Manuscript'.

We look forward to receiving your revised manuscript.

Kind regards,

Preeti Kanawjia, MD

Academic Editor

PLOS ONE

Journal Requirements:

Reviewers' comments:

Reviewer's Responses to Questions

**Comments to the Author**

1. Is the manuscript technically sound, and do the data support the conclusions?

Reviewer #1: Partly

Reviewer #2: Yes

2. Has the statistical analysis been performed appropriately and rigorously? 

Reviewer #1: I Don't Know

Reviewer #2: Yes

3. Have the authors made all data underlying the findings in their manuscript fully available?

Reviewer #1: Yes

Reviewer #2: Yes

4. Is the manuscript presented in an intelligible fashion and written in standard English?

Reviewer #1: Yes

Reviewer #2: Yes

5. Review Comments to the Author

Reviewer #1: The authors present an interesting study entitled "Validity of self-reported weight and height among female young adults in the United Arab Emirates". The study presents the findings on the validity of self-reported height and weight measures in a small group of female students within a defined age range. In its current form, the manuscript provides insufficient information for additional replication, and may be of limited utility given its small sample size.

Major issues:

1. Kindly provide the formula and/or calculation justifying why the sample size of 131 is sufficient for a study of this nature.

2. Kindly provide more details on the scale used for height measurements - specifying the make and model of the scale, and describe how it was set up to assure accuracy.

3. Line 158 states that Bland-Altman plots were performed to assess the agreement between self-reported and directly measured weight and height. Figure 1 only shows 1 Bland-Altman plot, comparing measured to self-reported weight. Please provide both the weight and height BA plots.

4. The conclusions of the study are incongruous with the reported results. The authors conclude that "The findings of this study propose that self-reported values should not be taken into account when determining the prevalence of unhealthy body weight for targeting weight loss or gain interventions. In general, female university students in the UAE struggled to provide accurate self-reports of their weight and height, leading to inaccurate BMI scores." In fact, the results show strong ICC values where calculated, and that the mean discrepancies between self-reported and measured height and weight were low (Table 4). For instance, a discrepancy of 0.5 - 1 kg in body weight is not out of the ordinary, given fluctuations in body weight throughout the day and during the menstrual cycle - particularly pertinent in this group of participants who were all female. Furthermore, discrepancies of less than 1 cm in height is negligible, and fall within the margin of error for height measurements. The discrepancies in the BA plot appear to be driven entirely by a small number of individuals, for whom there may be other factors leading to them providing inaccurate self-reported weight values. Kindly justify the conclusions made.

5. There appears to be substantial similarities between the text in this article and similar published articles. Please run this text through a plagiarism detection software and correct wherever appropriate.

6. There were no pre-defined range of acceptable limits of height and weight measure discrepancies, which should be defined a priori. Please refer to the following: https://www.ncbi.nlm.nih.gov/pmc/articles/PMC4470095/

7. Kindly justify the statement "However, if these values were to be used to determine the prevalence

368 of underweight or overweight, or target students for nutritional interventions programs, then

369 approximately one fifth of those who were underweight or overweight would be misclassified as

370 normal weight." 5 participants self-reported as normal but were underweight, whereas 7 overweight individuals self-reported as normal. 12 of 131 participants to not make up one fifth. Additionally, these findings should include the caveat that these findings were based on a very small number of individuals.

Minor issues:

1. SD was not fully defined at first mention.

2. Spelling error on line 158: testes in place of tests.

Reviewer #2: I would like to applaud the efforts of the authors for conducting a study to validate the self-reported measures of height and weight. Please see my comments below to improve the content and validity of the manuscript.

•Line 147 Add the hypothesis of the study

•Line 131-133 Is it a post-hoc analysis, then add it under the discussion section while discussing methodological considerations and limitations of the study

•Line 135 Any other contraindication for bioelectrical impedance analysis?

•Line 140 Explain more about “trained researchers” – their qualifications, specialization, and relevant experience

•Line 144 Add the scale/portable stadiometer details used for measuring the height of participants

•Line 147 Add the reliability and validity of the portable Tanita Body Composition Analyser (BC-420MA)

•Were the participants assessed at a fasting state in the morning using the portable Tanita Body Composition Analyser (BC-420MA)?

•Remove zero (dotted) line for the Bland-Altman plots

•Add a brief explanation of the absences of heteroscedasticity in the plots

•Add a linear regression analysis to explain the absence of a proportional bias in the plots

•Future recommendations should include similar studies on male participants

Thank you for the opportunity to review the manuscript

6. PLOS authors have the option to publish the peer review history of their article (what does this mean?). If published, this will include your full peer review and any attached files.

Reviewer #1: No

Reviewer #2: No

---

## [Author Response · Author response to Decision Letter 0]

13 Mar 2024

Editor Comments

Response: We have checked that the manuscript meets PLOS one’s style requirements.

Response: We have not deposited our dataset in a repository, however we would be happy to provide this upon request from readers.

Response: Not Applicable. All tables are embedded within the manuscript, and as per the PLOS one requirement, the figure has been uploaded separately which will then be embedded within the manuscript. 

Reviewers' comments:

Reviewer #1: The authors present an interesting study entitled "Validity of self-reported weight and height among female young adults in the United Arab Emirates". The study presents the findings on the validity of self-reported height and weight measures in a small group of female students within a defined age range. In its current form, the manuscript provides insufficient information for additional replication, and may be of limited utility given its small sample size.

Major issues:

1. Kindly provide the formula and/or calculation justifying why the sample size of 131 is sufficient for a study of this nature.

Response: As this was a post-hoc analysis, sample size calculation prior to conducting the study was not done. This information has been added to the manuscript. Furthermore, the calculation used to justify the sample size along with the references have been included in the manuscript.

2. Kindly provide more details on the scale used for height measurements - specifying the make and model of the scale, and describe how it was set up to assure accuracy.

Response: Information detailing the scale used for height measurements has been added.

3. Line 158 states that Bland-Altman plots were performed to assess the agreement between self-reported and directly measured weight and height. Figure 1 only shows 1 Bland-Altman plot, comparing measured to self-reported weight. Please provide both the weight and height BA plots.

Response: I think the reviewer has missed seeing the other 2 figures, as they were on different pages. The figure shows 3 bland-Altman plots (for weight, for height and for BMI. We have re-uploaded the BA plots into a different format document for clarity. 

4. The conclusions of the study are incongruous with the reported results. The authors conclude that "The findings of this study propose that self-reported values should not be taken into account when determining the prevalence of unhealthy body weight for targeting weight loss or gain interventions. In general, female university students in the UAE struggled to provide accurate self-reports of their weight and height, leading to inaccurate BMI scores." In fact, the results show strong ICC values where calculated, and that the mean discrepancies between self-reported and measured height and weight were low (Table 4). For instance, a discrepancy of 0.5 - 1 kg in body weight is not out of the ordinary, given fluctuations in body weight throughout the day and during the menstrual cycle - particularly pertinent in this group of participants who were all female. Furthermore, discrepancies of less than 1 cm in height is negligible, and fall within the margin of error for height measurements. The discrepancies in the BA plot appear to be driven entirely by a small number of individuals, for whom there may be other factors leading to them providing inaccurate self-reported weight values. Kindly justify the conclusions made.

Response: We have revised the conclusions of the study to be more in line with the reported findings of the study. 

5. There appears to be substantial similarities between the text in this article and similar published articles. Please run this text through a plagiarism detection software and correct wherever appropriate.

Response: We have run the text through a plagiarism detection software and have revised it accordingly.

6. There were no pre-defined range of acceptable limits of height and weight measure discrepancies, which should be defined a priori. Please refer to the following: https://www.ncbi.nlm.nih.gov/pmc/articles/PMC4470095/

Response: The Bland and Altman Plots illustrate differences between measured and self-reported measures of weight and height. Individual differences within 2SD of the mean are considered acceptable levels of errors. These have already been included in the manuscript.

Is the reviewer here referring to acceptable limits of measures that are of clinical significance? If so, we have not seen in the literature what would be considered acceptable. What we found are studies reporting differences in measured and self-reported- similar to what we have included on our study, and what we also found are studies that assess the reliability of height and weight measures between different observers. Furthermore, we found studies reporting weight fluctuations (over the week/seasonal etc.) and have added a reference to that in the discussion section. As there was no time lag of self-reported and measured, this limits the time for weight fluctuations and has been added as a strength of our study.

7. Kindly justify the statement "However, if these values were to be used to determine the prevalence

368 of underweight or overweight, or target students for nutritional interventions programs, then

369 approximately one fifth of those who were underweight or overweight would be misclassified as

370 normal weight." 5 participants self-reported as normal but were underweight, whereas 7 overweight individuals self-reported as normal. 12 of 131 participants to not make up one fifth. Additionally, these findings should include the caveat that these findings were based on a very small number of individuals.

Response: This is based on the sensitivity results obtained in relation to the underweight and overweight categories that were shown in table 5. For the underweight category it was 80.8% and for overweight category sensitivity was 76.7%. This means approximately 20% (or one fifth) would be classified incorrectly within each category. As this may have been confusing, we have now rephrased and clarified this finding in the discussion section to report on total sample that were misclassified (18 out of 131). The sample size has been added as a limitation to the study. Furthermore, the conclusions of the study have been revised accordingly.

Minor issues:

1. SD was not fully defined at first mention.

Response: This has been added.

2. Spelling error on line 158: testes in place of tests

Response: This has been corrected.

Reviewer #2: I would like to applaud the efforts of the authors for conducting a study to validate the self-reported measures of height and weight. Please see my comments below to improve the content and validity of the manuscript.

•Line 147 Add the hypothesis of the study

Response: The aim of the study has been added.

•Line 131-133 Is it a post-hoc analysis, then add it under the discussion section while discussing methodological considerations and limitations of the study

Response: This is a post-hoc analysis and has been added in the sample size calculation part. We have also added it as a limitation to the study.

•Line 135 Any other contraindication for bioelectrical impedance analysis?

Response: Yes, those who had medical implants were also excluded from the study. This has been added to the manuscript.

•Line 140 Explain more about “trained researchers” – their qualifications, specialization, and relevant experience

Response: Information on the qualifications of the researchers has been added to the manuscript.

•Line 144 Add the scale/portable stadiometer details used for measuring the height of participants

Response: Information on the stadiometer has been added.

•Line 147 Add the reliability and validity of the portable Tanita Body Composition Analyser (BC-420MA)

Response: For the purpose of this study, we have used the TANITA Body Composition Analyser only for the reporting of weight to the nearest 0.1kg, and not to assess body composition. Tanita has been reported to be reliable and valid in estimating body composition in epidemiological studies, However, a reference to that has not been added in our manuscript as body composition measures were not included in this study. 

•Were the participants assessed at a fasting state in the morning using the portable Tanita Body Composition Analyser (BC-420MA)?

Participants were not fasting when assessed. For the purpose of this study, body composition parameters (such as % fat, fat mass and fat-free mass) were not reported. The Tanita body composition analyser was used to report participants measured weight to the nearest 0.1KG on the same day participants self-reported their weight. Therefore, information on fasting was not included as it was irrelevant to the aim of the study. 

•Remove zero (dotted) line for the Bland-Altman plots

Response: The zero dotted lines have been removed from all the BA plots.

•Add a brief explanation of the absences of heteroscedasticity in the plots

Response: Regression analysis and a brief explanation was added to further explain this.

•Add a linear regression analysis to explain the absence of a proportional bias in the plots

Response: Linear regression analyses on the difference and mean of the measurements was conducted in investigate if there was a bias over the range of measurements. This was added to the results section of the manuscript.

•Future recommendations should include similar studies on male participants

Response: This has been added as a limitation to the study and a recommendation.

---

## [Decision Letter · Decision Letter 1]

4 Apr 2024

Validity of self-reported weight and height among female young adults in the United Arab Emirates

PONE-D-23-35645R1

Dear Dr. Haroun,

We’re pleased to inform you that your manuscript has been judged scientifically suitable for publication and will be formally accepted for publication once it meets all outstanding technical requirements.

Kind regards,

Preeti Kanawjia, MD

Academic Editor

PLOS ONE

Additional Editor Comments (optional):

Reviewers' comments:

Reviewer's Responses to Questions

**Comments to the Author**

1. If the authors have adequately addressed your comments raised in a previous round of review and you feel that this manuscript is now acceptable for publication, you may indicate that here to bypass the “Comments to the Author” section, enter your conflict of interest statement in the “Confidential to Editor” section, and submit your "Accept" recommendation.

Reviewer #1: All comments have been addressed

Reviewer #2: All comments have been addressed

2. Is the manuscript technically sound, and do the data support the conclusions?

Reviewer #1: Yes

Reviewer #2: Yes

3. Has the statistical analysis been performed appropriately and rigorously? 

Reviewer #1: I Don't Know

Reviewer #2: Yes

4. Have the authors made all data underlying the findings in their manuscript fully available?

Reviewer #1: Yes

Reviewer #2: Yes

5. Is the manuscript presented in an intelligible fashion and written in standard English?

Reviewer #1: Yes

Reviewer #2: Yes

6. Review Comments to the Author

Reviewer #1: The authors present an interesting study entitled "Validity of self-reported weight and height among female young adults in the United Arab Emirates". The study presents the findings on the validity of self-reported height and weight measures in a small group of female students within a defined age range. The authors have addressed my previous comments or acknowledged the limitations of their study in the text.

There remains several serious issues with the submission:

Major issues:

1. The resolution of the BA plots are of far too low resolution in the given file to be assessed. Only one file is provided to authors, in which the figures are embedded. Please ensure the figures provided to the journal are of high quality.

2. Limitations that should be stated and discussed are that the sample size was small, and that the observed group was of a limited demographic.

Minor errors:

1. Line 224: 0.976 was missing a decimal point.

2. Line 156: Are these undergraduate studies or do they already hold a Bachelor's degree in Public Health and Nutrition?

3. Figure 1 is not cross-referenced in the text.

Reviewer #2: (No Response)

7. PLOS authors have the option to publish the peer review history of their article (what does this mean?). If published, this will include your full peer review and any attached files.

Reviewer #1: No

Reviewer #2: No

---

## [Editor Report · Acceptance letter]

8 Apr 2024

PONE-D-23-35645R1 

PLOS ONE

Dear Dr. Haroun, 

I'm pleased to inform you that your manuscript has been deemed suitable for publication in PLOS ONE. Congratulations! Your manuscript is now being handed over to our production team.

Kind regards, 

on behalf of

Dr. Preeti Kanawjia 

Academic Editor

PLOS ONE